**Data Availability Statement:** Data can be found at the Open Science Framework (OSF): Persons with Disabilities- Disasters in Tuvalu. https://doi.org/10.17605/OSF.IO/JZ43D (Mohammadnezhad, 2020).

# Exploring persons with disabilities preparedness, perceptions and experiences of disasters in Tuvalu

**Natano Elisala[1], Amelia Turagabeci[2], Masoud Mohammadnezhad🔟[2]\*, Tamara Mangum[2]**

**1** Department of Public Health, Tuvalu Ministry of Health, Funafuti, Tuvalu, **2** School of Public Health and Primary Care, Fiji National University, Suva, Fiji

\* masoud.m@fnu.ac.fj

## Abstract

### Background

Historically, Persons with Disabilities (PwDs) are disproportionately affected by disasters. In Pacific Island Countries (PICs), the risks and vulnerabilities of PwDs arise from social inequalities, as well as environmental barriers. As the frequency and intensity of disasters will increase over the next decade, it is critical that the challenges faced by PwDs are addressed and that they are prepared.

### Objective

This study explores disaster preparedness, perceptions, and experiences with disasters among PwDs in Tuvalu.

### Methods

This qualitative study was carried out among people with physical and sensory disabilities and without post-traumatic stress disorder (PTSD), who are aged 21 and above. Using grounded theory methodology, semi-structured in-depth interviews were conducted with 24 participants, with 7 then participating in a Focus Group Discussion (FGD).

### Results

A total of 31 PwDs participated, of which 65% were male and 35% female, with a mean age of 44 ±15.70. The results showed how the PwDs ability to prepare, their perceptions, and experiences with disasters have contributed to their resilience to disasters.

### Conclusion

This study highlights the importance of understanding PwDs lived disaster experience to improve their preparedness and resilience for future disasters. This knowledge will assist government and non-government organisations, communities, and families to develop policies and plans that will enhance the preparedness of PwDs for disasters.

**Funding:** The author(s) received no specific funding for this work.

**Competing interests:** The authors have declared that no competing interests exist.

## Introduction

Disasters present a real challenge for persons with disabilities (PwDs), but strategies that prepare them for disasters are pivotal in mitigating health and social impact. These strategies include: addressing the vulnerabilities of PwDs living in disaster-prone countries; the use of PwDs' experiences and perceptions of prior disasters; addressing barriers to disaster preparedness; and preparing for disasters by the use of services offered to PwDs during disasters such as voluntary service, accessibility to communications and government/NGO rescue personnel [1,2]. Despite efforts in the area of disaster preparedness, often PwDs are left behind, overlooked, marginalised, and discriminated against [2–4].

The World Report on Disability states that 15% (est. 480 million) of the world's population are disabled persons, with 8% of that population living in developing countries [4,5]. The World Health Organization (WHO) estimates that 350 million people living with some form of disability globally are affected by natural disasters annually [6]. It is estimated that these numbers will increase due to ageing, lifestyle diseases, poverty, and accidents [7].

Historically, PwDs are disproportionately affected by disasters as they have various functional limitations and are more likely to be socially isolated and live in poverty [8]. With various functional limitations, their needs differ within the different types of disabilities and the degree of disability. For instance, a visually impaired person may respond by hearing about the warnings from television or radio of an impending disaster. This may be harder with a deaf and cognitively impaired individual to decide for themselves as they would rely solely on their carer to assist with the evacuation [8,9]. The factors causing vulnerability of PwDs to disasters are structural determinants, including social, political, economic, cultural, and environmental contexts which can influence the outcome of individuals when encountering disasters [2,9]. To reduce disaster risk and vulnerability amongst PwDs, the United Nations International Strategy for Disaster Risk Reduction (UNISDR) advocates risk reduction by the implementation of the Sendai Framework for Disaster Risk Reduction and other international agreements [10].

The Pacific Island Countries (PICs) are vastly dispersed in the Pacific Ocean, which covers thirty percent of the global surface [11]. These island countries are not only different in their geographic spread across the Pacific but also topographically different, being categorised into 'continental islands', 'oceanic islands', and 'raised limestone islands' [11]. Living in these islands can exacerbate PwDs vulnerability as these islands are highly exposed and frequently experience severe disasters [4,6,8,11,12]. According to the 2012 World Risk Report, PICs are among the most at-risk countries from disaster, which was based on exposures to hazard, susceptibility, lack of coping capacities, and adaptive capacities [1]. Surviving disasters in Tuvalu, a low-lying country with narrow coral atolls that sits barely three metres above sea level, can be a challenge for PwDs as many are unemployed, discriminated against, poorly educated, and frequently excluded from disaster management processes [2,13]. The vulnerabilities PwDs experience in the islands will increase as the frequency and intensity of disasters in the Pacific Island Countries will increase over the next decade [14,15].

There are limited studies on how PwDs experience changes to risk perceptions, with prior experience of a disaster playing a key role in determining preparedness [2,16–19]. For instance, PwDs were more responsive and prepared for disasters only after the 2010 earthquake experience in Canterbury, New Zealand [20]. Disaster experience of PwDs in evacuation centres provides valuable insight into construction that considers the availability of essential facilities such as toilets, social and support services, and understanding cultures and norms within societies [4,21–23]. Studies have shown that experience, education, and trust in authorities have a substantial impact on preparedness [24–26]. Furthermore, these perceptions are reinforced or modified through media reports, peer influences, and other communication processes [25,26].

The guiding principles of the Convention on the Rights of Persons with Disabilities (CRPD) stipulate non-discrimination, full and effective participation in society, respect of children and their rights, equality of opportunity, and accessibility [4,6]. However, globally, employment rates are among the lowest for PwDs with low socioeconomic status compared to their able-bodied counterparts [27–29]. As a consequence of being unemployed, accessing health and education can be difficult given that they cannot afford to pay their bus fares to hospitals, transportation, medical treatment, medicines, and inclusive education [4,6,8]. In the event of a disaster, emergency systems for alerting deaf individuals play a critical role in evacuation. The lack of information impacts the level of knowledge about disasters. This in turn increases PwDs vulnerability to disasters as they would not know what the disaster is and how to prevent or protect themselves from it [2,30,31]. The lack of assistance and information cost the lives of PwDs in Sri Lanka as they did not understand the information to evacuate during the 2015 Tsunami [9,28,32].

Another barrier affecting PwDs is the exclusion from decision making during disaster management planning and delivery. Specifically, they are rarely involved in disaster management planning despite inclusive advocacy, and guidelines on the inclusion of PwDs that were never implemented during the tsunami in Asia, including the process of post-tsunami reconstruction [2]. PwDs inclusion in disability-inclusive disaster risk reduction management practices at all stages of disaster will enhance their resilience, accessibility to information, and improve preparedness efforts for disasters and reduces stigmatization and discrimination [2,3,6,10].

Disaster preparedness involves establishing warning systems, mapping out evacuation routes, and educating families to maintain essential supplies and procedures to inform leaders of the communities in the event of a disaster [6,33,34]. Preparing PwDs can be influenced by knowledge, information, and communication. Knowledge plays a crucial role in reducing risk from disasters and is fundamental for informed decision making and the transfer of information. Studies have shown that there are clear benefits and advantages of both conventional scientific knowledge and traditional knowledge in making decisions about disaster preparedness and reducing risk to disasters [35–40]. Information and communication are effective tools in reaching out to the public and warning them of impending disasters. For PwDs, efforts to warn and prepare them requires effective risk communication strategies [41,42]; information that is accessible and understandable [12,42–45]; effective communication from trustworthy sources [41,46]; and the use of supportive policies and regulations [47,48]. The WHO and IFRC advocate specific methods of communicating with PwDs including audio signals, visual aids such as signs and gestures, and printed materials such as posters, leaflets, and pictures [6,49].

With the paucity of research on PwDs and disasters in Tuvalu and other PICs, this study has the potential to raise awareness and contribute to the body of knowledge of the challenges faced by PwDs living in developing countries during disasters. The evidence in this study will further provide knowledge in preparing and reducing PwDs risk from disasters as required by the Sendai Framework for Disaster Risk Reduction (SFDRR) [10]. The Sendai Framework for Disaster in its Guiding Principles in Paragraph 19(g) states the importance of using inclusive risk-formed data to reduce risk during disasters for PwDs.

## Methodology

### Study design sample selection

A qualitative grounded theory approach was chosen for the study. A qualitative research approach is a useful method for the phenomenon under investigation where the researcher tries to understand individuals' experiences and actions [50–52].

Purposive sampling was used to select 31 participants based on inclusion and exclusion criteria. PwDs were selected to participate if: they were 21 years and older; with physical or sensory impairment; do not exhibit post-traumatic stress disorder symptoms which were established by using a PTSD tool; and resides on the islands of Funafuti, Nukufetau, Vaitupu, and Nui. As some PwDs may have moved to Funafuti from Nanumea, Nuitao, and Nanumaga after TC Pam, they were also included in the study if they met the aforementioned criteria. PwDs were excluded from the study if they had intellectual disabilities or those that declined to participate, as outlined in Table 1. The goal of purposive sampling was to focus on particular characteristics of the interest population who were able to answer the research questions. In this study, PwDs were the target population, particularly those that had experienced TC Pam in 2015, as well as other disasters.

## Study sites and data collection tool

The tropical cyclone Pam in 2015 and other disasters in previous years have caused a great deal of destruction to buildings and properties. Depending on the severity of the cyclones, some islands were more affected than others and this study selects these islands particularly the central and northern islands to capture the experiences of PwDs [10]. Tuvalu is made up of 9 coral islands and the researcher travelled by government vessels to interview participants on four islands including Funafuti, Nukufetau, Vaitupu, and Nui. To facilitate proper interviews, the venues are important and outer island clinics were chosen as they provided a conducive environment for PwDs. Interviews on Funafuti were carried out in two venues: at the Disabled Persons Organisation Centre "Fusi Alofa," as the majority were familiar with the environment, and at homes for those participants who preferred interviews at their own homes.

Data was collected using semi-structured interviews and focus group discussions (FGD) from the 20th March to 10th April 2017. The interview questions were based on Pokras' [53] study on disaster preparedness and were further guided by the WHO Guidance Note on disability and emergency risk management for health research [6]. Moreover, to answer the study research questions the researchers' personal experience in preparing family members living with disabilities, and experiencing and surviving TC Pam, and experiencing all disasters occurred on the capital island in the past ten years have contributed to preparing relevant interview questions (see Table 2). These experiences were incorporated in the interview questions on how TC Pam affected PwDs, what were some of the barriers they experienced during TC Pam, and how to mainstream disability into community decision-making. Further, the focus group questions aspired to learn details and have in-depth discussions regarding PwDs disaster experiences, learning experiences, the barriers, and participation, not limiting their discussions to TC Pam but other past disasters, as well.

The interview questions were translated into Tuvaluan language by a bilingual translator for ease of administration by the researcher. The purpose of the translated questionnaires to the local language was to ensure participants understood the questions as well as giving

**Table 1. Inclusion and exclusion criteria.**

Inclusion criteria were:
 a) PwDs aged 21 years and above with physical or sensory impairment.
 b) PwDs, registered at the Ministry of Health, Ministry of Home Affairs, and Fusi Alofa Association.
 c) PwDs without posttraumatic stress disorder symptoms (PTSD), established by using the PTSD tool.
 d) PwDs residing in the islands of Funafuti, Nukufetau, Vaitupu, and Nui islands.
 e) PwDs from the islands of Nanumea, Nanumaga, and Niutao but reside on Funafuti island
 f) Experience TC Pam and other disasters
Exclusion criteria were:
 a) PwDs, with intellectual disability as defined by WHO [6]
 b) PwDs who were not willing to participate in this study

**Table 2. Questions for interviews and focus group discussions.**

| Interview Questions |
| --- |
| 1. What are some of the types of disasters that you have gone through, and can you share the experience? |
| 2. How did TC Pam affect you and your family? |
| 3. What were some of the barriers that prevented you from preparing for the disaster? |
| 4. In what ways do you think PwDs should be included and participate in decision-making in families, communities, and government? |

| Focus Group Discussions (FGD) |
| --- |
| 1. What are the emergencies that you believe we are at the highest risk for in this country? Why? |
| 2. If a disaster would occur next year, what would you change in your actions? How would you respond? |
| 3. Are you aware of any alert system that warns us that we are in danger of an emergency? |
| 4. Are you aware of any specific plans for emergencies here at Fusi Alofa or Government, Red Cross, or families at home? |
| 5. Which steps would you take if you heard an emergency warning? What would be the first thing that you would do? |
| 6. What would motivate you to prepare for an emergency and seek information on what to do before and during an emergency? |
| 7. Whom would you trust to talk to about emergencies? Why? |
| 8. If we wanted to share with the members of the Fusi Alofa or anyone with disabilities information about preparing for an emergency, what would be the best way to communicate the information? |

truthful responses. After the translation, a second independent translator back-translated to English and then further compared with the original. The researcher consulted with translators and worked on addressing the differences in translation and agreed on a translation that worked well with Tuvaluan language and context [54]. The translated questionnaires were then pretested on 5 participants including physically disabled and hard-of-hearing disabled persons. Participants from general populations were included in the pre-test to cater to carers and sign language users who would be involved in the actual data collection interviews. Comments from the participants were reviewed and incorporated in the final questionnaires.

## Study procedure

Each participant registered at the Ministry of Health, Ministry of Home Affairs, and Fusi Alofa was invited to the health clinic and given an information sheet and written consent form. The participant or carer signed the consent if they agreed to participate. For ethical considerations, participants who gave their consent were screened for PTSD symptoms by the nurse or health practitioner. It was critical that participants with PTSD were excluded, as recalling lived experiences can trigger terrifying events that could further debilitate their mental health conditions As such, those participants with symptoms were excluded while those without were further informed verbally regarding the study, how they were to contribute to the study and that all information provided would be kept strictly confidential. Participants on the capital with mobility issues were interviewed by the investigator at their homes and the rest of the participants were interviewed at the PwDs centre.

The participants in the interviews on the island of Funafuti were recruited to participate in the FGD. There were 31 interviewees of whom 21 had a physical impairment, 2 were blind and 8 were deaf or hard-of-hearing. Seven participated in the only FGD of which 5 had physical disabilities, 1 was blind and 1 deaf. There were challenges during the interviews, but a sign language translator assisted with the deaf person's interview, sitting close to the participant so they could hear properly, and visiting homes to interview those who had physical impairments to successfully collect data. Transportation was arranged for participants to travel to the disabled person's centre for the FGD. There were no FGDs on the islands of Nukufetau, Nui, and Vaitupu, as there were a limited number of PwDs on the islands.

The in-depth interviews started by greeting the participant and making them feel comfortable by talking about the weather. As soon as the participants had settled, the investigator introduced himself and the purpose of the study, which was the study on disability and their experience and clarifying the procedures of the interview. The participants were then informed that they could withdraw from the interview if they were uncomfortable with the questions. A series of questions were then asked based on the preparation guide and the participants were given ample time to respond to each question. They were not pressured to answer immediately, and this was particularly useful for participants who were deaf or hard of hearing, as an interpreter was used to convey the questions to the participant as well as interpreting to the researcher.

The FGD used the PwDs Centre computer room where the researcher greeted the participants, followed by the introduction of the study. As most participants were involved in the in-depth interview, the investigator briefly introduced the purpose of discussions followed by the discussion using a discussion guide. The investigator encouraged participants to share their views as there were no right or wrong answers. Their answers only reflect their individual experiences of disasters.

## Data collection

The interviews were recorded using a recorder, and the researcher used a notebook to take memos for each interview. The average time for the 31 interviews was 10.85 minutes with the longest being 20.19 minutes and the length of the FGD being 54.32 minutes. The interview data were transcribed verbatim on the night of each interview by the investigator. The transcribed data were then translated to English by a bilingual translator at the end of the final transcription of interview data for final analysis. These transcripts were then cross translated by another bilingual translator checking for consistency with the Tuvalu version. Any discrepancies in the translation to English were discussed between the translators and researcher and amendments were made accordingly.

## Data analysis

The data were analysed manually at the beginning of the process using individual interview transcripts. Each transcript was read for familiarity and coding followed thereafter. The codes were "constantly compared" during the process and according to Charmaz, [52] constant comparison of codes is the process of checking how codes or categories hold up against all interview data and previous data. If the codes were similar, then one can be used and the other dropped. All codes were important, and the researcher had to check and verify that it was frequently mentioned in the data to warrant its place in the final selection of themes. The transcripts were analysed using a thematic inductive analysis approach based on Grounded Theory, where categories and themes are cited directly as emerging themes from the data and not from preconceived hypotheses [50–52]. The next step was to search for themes within the code and review to ensure they represent views of the majority of participants. The themes were then defined, and categories developed to best capture the experiences of participants. Quotations were labelled based on the number of participants as P1, P2, etc.

## Researcher and training

The researcher carried out all fieldwork including interviews and facilitating the group discussions. The research was part of his master's thesis project. The researcher underwent training as an interviewer and data collector for a national STEPs survey in 2015.

**Table 3. Demographics characteristics of participants (n = 31).**

|  |  | Frequency | Percentage |
|---|---|---|---|
| Sex | Male | 22 | 65 |
|  | Female | 9 | 35 |
| Age (YR) | 21–35 | 11 | 36 |
|  | 36–55 | 13 | 42 |
|  | 56 and above | 7 | 22 |
| Marital Status | Single | 17 | 55 |
|  | Married | 10 | 32 |
|  | Divorce | 0 | 0 |
|  | Widow(er) | 2 | 6 |
| Employment Status | Employed | 2 | 6 |
|  | Unemployed | 29 | 94 |
| Income per month (AUD) | < $50 | 15 | 48 |
|  | $51–99 | 10 | 32 |
|  | >$100 | 4 | 13 |
| Type of Disability | Physical | 21 | 84 |
|  | Sensory: Blind | 2 | 6 |
|  | Sensory: Deaf/HH | 8 | 26 |

## Ethics approval

This study was approved by the Fiji National University College of Health Research and Ethics Committee (CHREC) and the Tuvalu Ministry of Health.

## Results

### Demographic characteristics of participants

The final sample selected and interviewed was 31, of which 65% were male and 35% were female. The age of participants ranged from 21 to 78 (*mean* = 44±15.70) with the majority not married and unemployed living on less than AU$50 per month and depending on family members for their wellbeing. The types of disabilities among the participants included physical, deaf, or hard of hearing and blind (Table 3).

### Themes

The themes to emerge from this study include capacity development, participation, communication, motivation, trusted sources, limitations of physical, hearing, and visual impairments and disaster experiences of different types of disabilities and PwDs experiences of different types of disasters. (See Table 4)

**Table 4. Categorises, themes, and sub-themes raised from the interviews.**

| CATEGORIES | THEMES |
|---|---|
| Factors contributing to | Capacity Development |
| Disaster Preparedness | Participation |
|  | Communication |
|  | Understanding barriers to disaster preparedness |
| Perceptions of preparedness | Motivation to prepare Trusted Sources |
| Experience | Disaster experiences of different types of disabilities |
|  | PwDs experiences of different types of disasters |

### Factors contributing to disaster preparedness

**1. Capacity development.** *'Capacity development'* is an important component for preparedness. Participants were requested frequent live evacuation drills and development of evacuation plans for PwDs.

*"There should be workshops, awareness programs, and training on disaster preparedness, especially for us disabled people to be inclusive."* [P14].

Another participant noted previously before TC Pam, PwDs would want to know or prepare for disasters, but since TC Pam has to be done for their safety given the impact disasters may have on their lives.

*"As we usually know that these things we had experienced have never been heard of (referring to TC Pam effects). So, some people don't take it to the task to prepare for disasters. But I want to thank the government and Fusi Alofa that they have a plan for PwDs. Plans for the Fusi Alofa, this office would be a meeting centre awaiting evacuation by the government. So, when the government decides what to do with PwDs, they can come to one place where they would extract them to a safe place."* [P14].

Despite concerns about PwDs not wanting to prepare before TC Pam and how little government departments recognise and include PwDs in decision making, participants suggested more awareness programs.

*"Awareness programs should include radio programs, posters, and other IEC materials including visiting government departments lobbying for inclusive policies as well as selling of handicrafts."* [P1-FGD].

**2. Participation.** Participation in decision making for disaster preparedness was mentioned by participants. They were aware of the importance of being involved in community and government decision-making bodies such as the disaster management department. The Government of Tuvalu in 2014 ratified the United Nations Convention on the Rights of Persons with Disabilities.

*"I think the way to include PwDs to participate is through the Convention on the Rights of Persons with Disabilities (CRPD). By protecting PwDs rights. At the moment there are so many associations and government departments we have gone to promote awareness of PwDs."* [P1].

Despite this, participants noted that the government needed to do more for PwDs. The participant clarified that although CRPD has been ratified, it needs implementation:

*"Because the government has ratified the Convention on the Rights of PwDs, now it's time to implement because, now it's time to implement because at the moment it has not been implemented. For example, the Ministry of Education plans for inclusive education, but they don't practice it. So PwDs should inform the government our needs because it's stipulated in the convention that PwDs should be involved in mainstream education. So that is what we should do this year."* [P7- FGD].

**3. Communication.** Communicating information to PwDs is critical before, during, and after a disaster. Most participants noted the various methods that can be used to effectively reach out to these individuals such as the caregiver, proximity of announcement, language and communication, and the use of sign language. Several participants expressed their belief that information that is communicated to PwDs is specific and should be delivered by their caregiver considering their relationship and disabilities, particularly those who are physically impaired, deaf, or blind. Wrong information or not communicated in the right manner would cause anxiety and confusion to PwDs.

*"So the messages can be related to those that are deaf during strong winds and this include looking straight at them and relate the message. Its' important that they can see our lips."* [P16].

However, even in the best interest of carers and families, different times of the day may cater to different sign language and a participant commented on an appropriate sign language method for deaf or hearing disabilities.

*"For deaf individuals, it's good during the daytime because we can use sign language. . .In my opinion, it's better to prepare signs such as specifically for waves (tsunami). If there's a tsunami, then use a sign specifically for waves such as colour picture of waves so that the deaf individuals would know of the disaster. If it's at night, then we should use a light so that deaf individuals would see and know that there is a wave or that the strong winds have picked up speed. And for the fire, well everyone sees the fire."* [P7].

**4. Understanding barriers to disaster preparedness.** Understanding and addressing the barriers to disaster preparedness would assist families and the government to prepare PwDs for disasters. Participants highlighted barriers to not having individual/family disaster/evacuation plans, unemployed, and inability to evacuate without assistance. Some of the participants mentioned that their families did not have an evacuation plan and neither do they know what the government plans are for PwDs.

*"I do not know whether Fusi Alofa or the Government have emergency plan. My family does not have a specific plan of preparation for such events."* [P28].

*"I am not aware of any plans for emergencies, but my husband and I know that if we are warned of a crisis like strong winds, then we will follow those instructions."* [P26].

Participants felt that preparing for disaster is a challenging issue since they cannot be ready in time before a disaster as they are unemployed. Data has shown that 90% are unemployed.

*"My husband and I do not work and the benefits from Fusi Alofa help us financially."* [P11].

One of the most challenging issues for most participants was the evacuation was most noted that they would need assistance.

*"I highly doubt there is anything I can do about that. The only thing I can depend on to help me during such events is my family and their effort to take me to a safer place. Therefore I think that there is nothing I could do about things like that except depending on my family."* [P29].

One participant gave an insight into why the public, including family members, need to know that PwDs limitations differ. For example, during disaster evacuation, persons with physical impairment may be assisted differently than those with hearing impairment.

*"As we know we have different disabilities they (rescuers) should know how to handle each one, if they don't know then it's a big problem."* [P2].

## Perception of preparedness

**1. Motivation to prepare.** Participants felt that motivation was an important factor in their preparations for disasters. Motivation is a purpose for them *'to stay alive'* for their families.

*"The Community Announcement Alert System (locally known as Te Valo), they announce close to every house about the strong wind. . .Sometimes I hear the announcements and sometimes I don't."* [P3].

Moreover, their purpose to continue their motivation is strengthened by the local alert system in Tuvalu called 'Te Valo'.

*"On this island, we usually hear the announcements on the radio. Previously there was a strong wind, I was still working and there was a Valo but we did not hear it from the radio, perhaps it was sent directly from Funafuti. But most of the time we hear announcements from the radio for people to prepare for strong winds that will hit (the island) at that time. . .Therefore, people are given ample time to prepare while it's still making its way."* [P1].

They are also motivated by their 'faith in God' that protects them and their families. They noted that their risks of being affected increases during disasters and therefore they prepare emergency food supplies, plan to relocate to the centre of the island or further from the sea early, build resilient infrastructure, are protected and dependent on families and communities, and pray to God as a coping strategy.

*"The first thing that we all need to do is pray because God is our help. Everything that we worry about, we should ask His help. We should always pray before we move to other places, whether with our families or anyone else."* [P25].

**2. Trusted sources.** '*Trusted Sources'* to the PwDs is important. The information must come from a trusted source, either the national government or local government, and needs to be related to PwDs in a manner they understand.

*"In my opinion, the weather people are trustworthy because they are the ones who observe the changes in the weather and know about disasters before they occur using their equipment. And also, I trust the disaster department, as they prepare the community for threats like this. Those are the people that I trust."* [P17].

*"Okay, I believe local announcements, because it's a message from the government to prepare all for disasters such as strong winds at that time. And at times local calls are not trusted, because when they announced but the waves have passed us already. But the important thing for the family is to prepare for emergencies. Because we do not know when it happens, but if*

*we're ready then we are ready whenever disaster strikes. Whether the calls or announcement of an emergency happens or not we should be ready."* [P2].

## Experience

**1. Disaster experiences of different types of disabilities.** *'Lived experiences'* is described as the actual personal account that an individual encountered living within a group. In this context, we define lived experience as personal experiences of PwDs living in Tuvalu. Personal disaster experiences of PwDs differ for each type of disability. Participants were those with physical, hearing, and visual disabilities.

For individuals with physical activity, the majority stated that they were vulnerable to strong winds, storm surges, and drought. Those with visual impairments mentioned that they were vulnerable to strong winds, drought, and storm surges, whereas for individuals with hearing impairments perceived that strong winds and storm surges were disasters that could impact their lives.

Participants with physical disabilities shared their experience and the majority felt that they were vulnerable to strong winds. Another participant perceived he was vulnerable to storm surge and shared his experience of TC Pam.

For physical disability participants they noted that:

*"..strong winds caused extensive damages to my house sometimes during midnight hours because at that time (TC Ula) I couldn't sleep those early morning hours because of the sound of the winds. . .I was scared because our house was far from the main settlement and I went to stay at my neighbours' house until dawn."* [P4—FGD].

*"At that time of the storm surges (TC Pam), I heard the community emergency alert where we lived and the waves were hitting our areas but the biggest hit was at the village side. A lady and her disability son's house was destroyed by waves. . .And was there assistance at that time? Those are the problems we faced at that time because everyone was busy and no one helped us, no one from Red Cross and no one from the Police department. If they helped there wouldn't be any difficulties faced. But when I saw the waves it was getting bigger, (sighed.), I was really worried."* [P14—FGD].

Participants with hearing disabilities have their own experiences and the common experiences of participants with hearing disabilities were that they did not know what was happening around them during a disaster.

*"I recall in one of my experience, people ran during a strong wind incident, but I wasn't afraid. It is only recently during the aftermath of TC Pam that I came to realise the reality of things especially on the island of Nui. Now I am afraid and I want more information provided in advance before any disaster happens."* [P4].

*"I didn't hear the announcement about storm surges [TC Pam], I only found out when I was at the local kitchen that's when the sea waters started flowing to the small hut where I was sitting. My small cabbage garden and my pig pen were destroyed at the time."* [P3].

Participants with visual disabilities also have different experiences during disasters.

*"..because of my disability (visual), I can be extremely affected by all these disasters (strong winds, storm surges, drought, and fire). Especially nowadays where houses are built with*

*corrugated roofs and they can be blown from roof tops quickly and can hurt me. As well as waves where I do not have any opportunity as it hits us I can also be extremely affected including fire where I can also be affected."* [P1].

*"Storm surges in the past have also affected our homes... We often just stay indoors to look out for water that might wash up into our house. Luckily for us, this never occurred as the sea-water drifts outside of our house."* [P26].

**2. PwDs experiences of different types of disaster.** The experiences of PwDs of different types of disasters differed according to participants. In general, most participants were aware of their limitations during any disasters and their reliance on family members or the public to assist them to evacuate to evacuation centres. There are three disasters frequently mentioned by participants that they are not only vulnerable to but have personal experience which includes strong winds, storm surges, and drought.

## Strong winds

Strong winds are seen as cyclones that have a devastating impact on the lives of people on the islands and participants knew these would occur during the year. During the cyclone season, many families prepare for the worst.

*"Strong winds occur close to the end of the year and beginning of the year and those are the seasons from strong winds."* [P17].

*"What do we prepare during cyclones, perhaps the usual. To prepare homes to be safe... secure areas of the house that are not well secure and the communities offers to help all those who need it at the time."* [P22].

Another participant with physical disabilities who witnessed severe tropical cyclone, Bebe, as one of the worse cyclones that have hit Tuvalu noted:

*"I was there during the strong winds [cyclone Bebe] and I witnessed it, if the waves are bigger than those that hit Nui (TC Pam) then I would not be able to do anything to survive."* [P6—FGD].

A vision-impaired participant gave an insight into how she felt during cyclone Bebe:

*"When I was little, there was once a hurricane that hit Funafuti I was only 7 years old I was living in the islands and the seas were rough at that time. But because I was young I didn't think like an adult than as I was still a child... Only later on in life or even now that I came to realise that there is an association of people with disabilities. So I now settle with thoughts of a disabled person. So from me, there should always be preparations so that we (disable people) are ready so that the government should be ready with a plan to save us people with disabilities."* [P4—FGD].

Similarly, for persons with hearing impairment, a cyclone threatens their livelihoods.

*"I think it's the sea level rising that I am affected by it because I live right next to the sea and it has a very big impact on our house. Their house is right near the sea, so when the wind blows (cyclone)... the waves come, at one time it hit my house.* [P5].

## Storm surges

Storm surges were experienced by some participants particularly those from the island of Niutao and Nui who identified the disaster as a depressing situation for everyone.

A person with a physical disability described how he felt during TC Pam and also witnessed a mother and her disabled sons' home destroyed by the storm surge:

> *"I was there at the time TC Pam that hit my island and what I felt very depressed about the situation. . .and I met at that time a lady and her disabled son whose home was destroyed"* [P1—FGD].

Participants with hearing impairment described her actions if there was a storm surge. They would prepare and store all necessary items in a bucket and the bucket serves also as a flotation device. Similarly, a person with visual impairment described how she would use a bucket during strong winds. This suggests that PwDs knew what they need to prepare based on experience and advice from authorities.

> *"When strong winds are announced, I prepare clothes for me and my granddaughter and torch and put them in the bucket together with lighter, and biscuit bucket, and a bottle of water. And my family always laughs at me when I prepare these things. I prepare these important items because if the disaster lasts longer the food prepared will help keep us healthy. I can also use dried clothes in the bucket after the disaster."* [P18].

## Drought

During the drought season, water becomes so precious that families ration its use for family members. Tuvalu solely depends on rainwater and its proper storage. Those with limited or no storage tanks are usually the first to suffer from water shortage. A physically impaired participant described his concerns during drought:

> *"Days with no water tend to have us all looking for water. This type of disaster especially affects me due to my body weight which makes it hard for me to search for water and to be clean at all times."* [P28].

Participants with hearing and visual impairment noted that drought is dangerous because it happens every year. Preparing is critical for their survival and they would always depend on the government and Red Cross to help them during these situations.

> *"I think that is what Fusi Alofa would like to see from the government that is to see that specific plans for people with disabilities during preparations and emergencies are developed and implemented."* [P1].

## Discussion

The objective of this study was to gain an understanding of PwDs concerning disaster preparedness, their perceptions, and their different types of experiences of disasters and experiences of different types of disasters.

This research identified factors contributing to disaster preparedness for PwDs as a relevant category, appropriately describing the theme capacity development among PwDs and the

community; participation in government and community decision-making process; and communication and understanding barriers that would impede preparedness. The findings concur with international agreements and studies that show how governments should involve PwDs in developing capacities of PwDs and disability organisations, as well as developing inclusive disaster preparedness programs [10,55,56]. These findings are important for PwDs so that governments recognise and develop strategies to assist in preparing them for disasters.

Capacity development was important to PwDs, particularly when it comes to evacuation. Evacuation drills increased knowledge within PwDs as well as the communities assisting them during evacuation. However, this study found that the abled-bodied training participants needed an in-depth understanding of different types of disabilities. During disaster evacuation, PwDs have different needs and evacuation teams need to understand them so that they don't get injured in the process. The Sendai Framework advocates that DRR activities such as evacuation drills are required to empower people who are disproportionately affected by disasters including the disabled and the poor [10].

Another factor that contributes to the preparedness of PwDs is their participation in communities, government, and NGO policy development and decision-making. Participants in this study shared their stories about how they are now participating in government policy developments and community meetings, but more advocacy is needed. The finding is consistent with studies showing that participating in community decision making empowers PwDs to increase their chances of preparing for disasters [6,8]. The International Handicap also supports this finding, noting that PwDs should participate in decision-making at all stages of humanitarian response during and after disasters so that authorities would understand their needs [56].

PwDs are a heterogeneous group that requires different disability communication approaches and correct methods of communicating information determine their understanding and ability to evacuate in time [49]. This present study showed that methods to deliver the message to PwDs is critical and proposes methods to reach out to these individuals. These methods consider communication with the caregiver, proximity of announcements, use of local language, and sign language. These results are consistent with studies on risk communication with disabled communities [6,12,43,49].

Participants in this study identified the following barriers that hinder preparedness efforts: lack of planning to prepare for disasters, both for families and government or NGO; unemployment; and needing assistance to evacuate in the event of a disaster. The findings from this study are comparable with the current literature in reports and studies on barriers to disaster preparedness [6,55,57,58]. The types of barriers commonly experienced by PwDs include attitudinal, communication, physical, policy, programmatic, social, and transportation issues [6,57,58]. Studies in the Pacific Islands identified that PwDs, specifically children and youth, are marginalised within their communities and the majority are unemployed, making it hard to buy necessities [7]. Moreover, marginalisation and discrimination may not improve PwDs participation in community decision-making and will consequently affect their preparations for disasters.

Understanding the role of unemployment in PwDs during a disaster is crucial. This study found that over 90% of participants were unemployed, which is consistent with reports on the PwDs employment rate [6,10,59,60]. Studies confirmed that PwDs have a consistently low socioeconomic status (low SES), are mostly unemployed, are less educated, and face health and financial challenges [2,61,62]. Without financial resources, PwDs are not able to buy food, water, and other essential items to prepare for disasters. Studies have also confirmed that by developing policies to protect the rights of PwDs, improving their job opportunities, and prioritising and responding to their needs will address inclusivity issues among general populations [5,58,63,64].

This study has also shown that PwDs require assistance to evacuate during disasters. Preparing PwDs would require families as first responders to disasters, as well as government and NGO assistance. The result is consistent with studies of individuals with physical disabilities requiring assistance to evacuate [6,65]. Most of the participants in this present study stated that they would need assistance during disasters and these perceptions are somewhat accurate, given that most of them asserted that during disasters, 'what else can they do' but to ask for assistance. Dunn found that the reliance of PwDs on assistance was a barrier, which is similar to the findings of this study [31].

This study found three factors for motivation to prepare: to stay alive; Te Valo (local alert system); and faith in God. Pacific Island culture and traditions have significant mechanisms to assist in preparing communities and families including disabled people for disasters [7]. Motivation to prepare is affected by how one views the risks of disasters [10]. Given the limited studies on PwDs and their behaviour, our finding is only consistent with general population studies, noting that they were motivated because of family safety, knowledge, information, and praying as a coping strategy [53,66,67].

This study found various sources of information that are trustworthy such as government departments including the police, weather, and the community alert system. Studies have shown that trusted sources are an important component of disaster preparedness [16,46]. Trust is diminished when the delivery of a message is delayed, as noted by this study. Nevertheless, by trusting the sources of information on impending disasters and appropriate communication with PwDs contributes to the preparedness and mitigation of disaster effects.

Scientific and traditional knowledge are both important factors in preparing for disasters. This study has shown that participants' perceived knowledge, concerning knowing disaster types, responding to and preparing for disasters, were contributing factors to disaster preparedness for PwDs. These findings are broadly consistent with current literature, where knowledge was seen as a key dimension or desired end-state of preparedness activities [35–37,68–71].

The findings from this study have shown that experiences are important factors in disaster preparations. According to the data, those with different types of disability perceived disasters are prone to differ from each other. The participants' reactions to disasters in this study are consistent with studies on the influence of experience in the formation of risk perceptions and disaster preparedness strategies [17,72,73]. Experiencing an injury can also increase the perception of vulnerability [74]. PwDs are a diverse group, and as such, meeting their needs is not as easy as applying a one-size-fits-all model, but rather requires tailored assistance for different disabilities [6,8].

The findings from experiences of PwDs showed how they use the knowledge of different types of disasters to prepare for disaster. Studies with similar findings found how experiences in disasters have influenced how they used that knowledge to help others, as well as developing preparedness plans [6,16,68]. This study has shown that the experiences of PwDs about disasters must not be neglected but rather encouraged in terms of the participation and inclusion of PwDs in decision and policymaking.

## Conclusion

Persons with disabilities are amongst the most vulnerable segment within the general population. They must receive continuous support to prepare for disasters. This study showed factors that contribute to the preparedness of PwDs in disasters, perspectives on preparedness and understanding lived experience in disasters. The factors contributing to their preparedness are capacity development in an evacuation, participation in government and Non-government

organisations and community decision-making, and the use of suitable methods of communi-cation. Understanding barriers that hinder these preparations are illustrated by not having a disaster plan, the impact of being unemployed, and the need for assistance to evacuate. Per-ceiving risk as a preparedness measure for disasters also motivates PwDs to stay alive for fami-lies, use trusted sources of information, and trust in God. Lived experiences brought a variety of valuable perspectives on different types of disabilities and types of disaster experiences. Fur-ther studies are recommended in the areas of policies and socioeconomic analysis, using phenomenological approaches to describe the meaning of lived experiences of PwDs and how tacit knowledge can be used in decision-making. This research suggests that government, Non-government organisations, communities, and families maintain assistance in preparing PwDs for disasters by addressing the factors that contribute to preparedness and the barriers and challenges.

## Acknowledgments

We would like to thank all the participants for taking out their valuable time and participating in the interview.

## Author Contributions

**Conceptualization:** Natano Elisala, Amelia Turagabeci, Tamara Mangum.

**Data curation:** Natano Elisala.

**Formal analysis:** Natano Elisala.

**Funding acquisition:** Natano Elisala.

**Investigation:** Natano Elisala.

**Methodology:** Natano Elisala, Masoud Mohammadnezhad.

**Project administration:** Natano Elisala.

**Resources:** Natano Elisala.

**Software:** Natano Elisala.

**Supervision:** Amelia Turagabeci, Masoud Mohammadnezhad, Tamara Mangum.

**Validation:** Natano Elisala.

**Writing – original draft:** Natano Elisala.

**Writing – review & editing:** Amelia Turagabeci, Masoud Mohammadnezhad, Tamara Mangum.

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
