## [Decision Letter · Decision Letter 0]

9 Sep 2020

PONE-D-20-18521

Exploring Persons with Disabilities Preparedness, Perceptions and Experiences of Natural Disasters in Tuvalu

PLOS ONE

Dear Dr. Mohammadnezhad,

Thank you for submitting your manuscript to PLOS ONE. After careful consideration, we feel that it has merit but does not fully meet PLOS ONE’s publication criteria as it currently stands. Therefore, we invite you to submit a revised version of the manuscript that addresses the points raised during the review process.

**The judgment of the three Reviewers was very positive. Basically, they mainly highlight shortcomings in the writing that require revision before the manuscript can be published. The Reviewers provided suggestions for the correction of certain expressions and phrases. But, if necessary, the Authors would do well to use a professional proofreading service.**

We look forward to receiving your revised manuscript.

Kind regards,

Stefano Federici, Ph.D.

Academic Editor

PLOS ONE

Journal Requirements:

2. When reporting the results of qualitative research, we suggest consulting the COREQ guidelines: http://intqhc.oxfordjournals.org/content/19/6/349. In this case, please consider including more information on the number of interviewers, their training and characteristics.

3. Please provide additional details regarding participant consent. In the ethics statement in the Methods and online submission information, please ensure that you have specified (1) whether consent was informed and (2) what type you obtained (for instance, written or verbal). If your study included minors, state whether you obtained consent from parents or guardians. If the need for consent was waived by the ethics committee, please include this information.

4.We note that you have indicated that data from this study are available upon request. PLOS only allows data to be available upon request if there are legal or ethical restrictions on sharing data publicly. For information on unacceptable data access restrictions, please see http://journals.plos.org/plosone/s/data-availability#loc-unacceptable-data-access-restrictions.

5.Please amend your authorship list in your manuscript file to include author Tamara Mangum.

Additional Editor Comments (if provided):

The judgment of the three Reviewers was very positive. Basically, they mainly highlight shortcomings in the writing that require revision before the manuscript can be published. The Reviewers provided suggestions for the correction of certain expressions and phrases. But, if necessary, the Authors would do well to use a professional proofreading service.

Reviewers' comments:

Reviewer's Responses to Questions

**Comments to the Author**

1. Is the manuscript technically sound, and do the data support the conclusions?

Reviewer #1: Yes

Reviewer #2: Yes

Reviewer #3: Yes

2. Has the statistical analysis been performed appropriately and rigorously? 

Reviewer #1: Yes

Reviewer #2: Yes

Reviewer #3: N/A

3. Have the authors made all data underlying the findings in their manuscript fully available?

Reviewer #1: Yes

Reviewer #2: Yes

Reviewer #3: Yes

4. Is the manuscript presented in an intelligible fashion and written in standard English?

Reviewer #1: Yes

Reviewer #2: Yes

Reviewer #3: Yes

5. Review Comments to the Author

Reviewer #1: In relation to question 3 above i assume all data are held by the appropriate departments who employ the authors. presumably the data are also confidential.

in relation to question 2 above, the surveyed population was small in size, but acceptable. More sophisticated statistics would not have enhanced the analysis.

In relation to question 4 above, the manuscript is intelligible but there are many mistakes. I have converted the pdf to Word and used track changes to suggest recommended corrections to the English expression. I have also inserted a number of review comments on specific statements. This copy of the paper is uploaded for your guidance in making corrections.

The results section could benefit from some reduction in length. It is a bit tedious and repetitive in several places.

The discussion section is very clearly stated and well connected to findings from the general literature.

Reviewer #2: This study investigates an understudied geographical area and vulnerable group of individuals with disabilities. The manuscript, after the revisions and addressing the comments from the previous reviewers, is sound and the data support the conclusion.

Please make adjustment to Focus Group Discussion and its abbreviation FDG. You may change it to FGD to be consistent with the order of words.

[Focus Group Discussion (FDG) in Abstract and Focus Group Discussions (FDG) in Table 2]

Reviewer #3: I have read the MS in depth and also the previous reviewers' comments and author responses. The authors' changes appear to well address the reviewers comments and i feel the MS is now a topical, socially valuable, and methodologically sound piece of work meriting publication. Articulating PwDs lived experiences of disasters and their suggestions for future disaster planning is very valuable for improving future disaster response/management strategies for this group. There are still some english and small grammatical issues that need attending to,

Suggestions;

P 2. Abstract: Conclusion, first sentence needs reworking to clarify meaning. "This study highlights the salience in understanding PwDs prepare and experience, which influences their resilience to disasters". Maybe change to something like "This study highlights the importance of understanding PwDs' lived disaster experience to improve their preparedness and resilience for future disasters".

P 6. Line 5 in Methodology. Insert sample size (31) "Purposive sampling was used to select "31" participants..."

P 8. Line 8. insert apostrophe after "researcher". "....the researcher's personal experience...."

P 9. Line 3. Change "participants understanding" to "participants understood"

P 9. Line 4. Change "translator translated back to english.." to "...translator back-translated to english.."

P 9. Lines 9 & 15. Change "users that" and participants that" to "users who... and "participants who..."

P 10. line 7. Change "interviews..." to "interviewees..."

P 11. Line 3. Change heading "Data management and analysis" to "Data Collection"

P 11. Insert heading ""Data Analysis" between lines 11 & 12.

P 13. Line 1. Change "The themes from this study..." to "The themes to emerge from this study..."

P 14. Lines 1 to 3. Delete ""For instance a participant asserted that there should be workshops and have more awareness programs, trainings on disaster.... to be inclusive". This sentence is repeated in entirety in italicized example immediately following.

P 15. Lines 10 & 11. Insert "United Nations" before "Convention on the rights of PwDs..." Also capitalize "R"ights of "P"ersons with "D"isabilities"

6. PLOS authors have the option to publish the peer review history of their article (what does this mean?). If published, this will include your full peer review and any attached files.

Reviewer #1: No

Reviewer #2: No

Reviewer #3: No

---

## [Author Response · Author response to Decision Letter 0]

23 Sep 2020

Academic Editor

Comment: Ensure the manuscript meets PLOS ONE style requirement.

Response: Thank you for your comment. We have revised the manuscript to meet the journal's style requirement including the naming of file.

Comment: When reporting results of qualitative research, suggest consulting COREZ guidelines. In this case, please consider including more information on the number of interviewers, their training, and their characteristics.

Response: Thank you for raising this important point. We have included a new section under ‘Researcher and training’ under methodology. This section contains information regarding the researcher being the interviewer, gender, and their experience and training. 

Reviewer 1.

Comment: In relation to question 4 above, the manuscript is intelligible but there are many mistakes. I have converted the pdf to word and used track changes to suggest recommended corrections to the English expression. I have also inserted a number of review comments on specific statements. This copy of the paper is uploaded for your guidance in making corrections

Response: Thank you for your comments and the corrections which is much appreciated. We have made all corrections according to your comments and further proofread and edited by one of the authors. All corrections are highlighted in yellow with comments on up to 7 points raised. For example Corrections to Para2, Line 2: “..without physical or sensory” to “ with physical or sensory impairments”.

Comment: The results section could benefit from some reduction in length. It is a bit tedious and repetitive in several places.

Response: Thank you for pointing this out. The authors have edited the results sections addressing repetitions and providing a clearer picture of PwD experiences. 

Reviewer 2. 

Comment: The study investigates an understudied geographical area and vulnerable group of individuals with disabilities. The manuscript, after the revisions and addressing the comments from the previous reviewers, is sound and data supports the conclusion. Please make adjustments to Focus Group Discussion and its abbreviation FDG. You may change it to FGD to be consistent with the order or words. [Focus Group Discussion (FGD) in Abstract and Focus Group Discussions (FDG) in Table 2]. 

Response: Thank you for your comments. The authors have adjusted FDG to FGD in the Abstract and in Table 2 as well as in all sections. 

Reviewer 3

Comment: There are still some English and small grammatical issues that need attending to:

Suggestion: P1 Abstract: Conclusion, the first sentence needs reworking to clarify meaning: “This study highlights the salience in understanding PwDs prepare and experience, which influences their resilience to disasters”. Maybe change to something like: “This study highlights the importance of understanding PwDs lived disaster experience to improve their preparedness and resilience for future disasters.”

Response: Thank you for your suggestions. The authors have agreed and changed P1 according to your suggestion.

Suggestion: P6 Line 5 in Methodology. Insert sample size (31). “Purposive sampling was used to select “31” participants…

Response: Thank you for the suggestion. The authors agree and have made an addition to the sentence which now read “Purposive sampling was used to select 31 participants based on inclusion and exclusion criteria.”

Suggestion: P 8 Line 8. Insert apostrophe after “researcher”. “..the researcher’s personal experience…”

Response: Thank you for the suggestion. The authors have inserted the apostrophe as suggested and the sentence now reads: “Moreover, to answer the study research questions the researchers' personal experience in preparing family members living with disabilities, and experiencing and surviving TC Pam, and experiencing all disasters occurred on the capital island in the past ten years have contributed to preparing relevant interview questions (see table 2).

Suggestion: P9. Line 3. Change “participants understanding” to “participants understood”

Response: Thank you for the suggestion. The authors agree and changed accordingly and the sentence now reads “The purpose of the translated questionnaires to the local language was to ensure participants understood the questions as well as giving truthful responses.”

Suggestion: P9. Line 4. Change “translator translated back to English..” to “..translator back-translated to English..”

Response: Thank you for your suggestion. The authors agree to your suggestion and have changed accordingly and the sentence now reads: “After the translation, a second independent translator back-translated to English and then further compared with the original.”

Suggestion: P9. Lines 9 & 15. Change “users that” and “participants that to “users who…and “participants who..”

Response: Thank you for the suggestion. The authors agreed and changed accordingly. The sentences read: L9 “Participants from general populations were included in the pre-test to cater to carers and sign language users who would be involved in the actual data collection interviews.” L15 “For ethical considerations, participants who gave their consent were screened for PTSD symptoms by the nurse or health practitioner.”

Suggestion: P10. Line 7. Change “interviews..” to “interviewees…”

Response: Thank you for the suggestion. The authors agree and the sentence now “There were 31 interviewees of whom 21 had a physical impairment, 2 were blind and 8 were deaf or hard-of-hearing”

Suggestion: P11. Line 3. Change the heading “Data management and analysis” to “Data Collection”

Response: Thank you for the suggestion. The authors agreed to the changes in the headings.

Suggestion: P11. Insert heading “Data Analysis” between lines 11 and 12.

Response: Thank you for the suggestions. The authors agree to the changes with the heading ‘Data Analysis” between lines 11 and 12.

Suggestion: P13. Line 1. Change “The themes from this study..” to “The themes to emerge from the study..”

Response: Thank you for the suggestion. The authors agree with the suggestion and the sentence now reads: “The themes to emerge from this study include capacity development, participation, communication, motivation, trusted sources, limitations of physical, hearing, and visual impairments and disaster experiences of different types of disabilities and PwDs experiences of different types of disasters.”

Suggestions: P14. Line2 1 to 3. Delete “For instance, a participant asserted that there should be workshops and have more awareness programs, training on disasters…to be inclusive” This sentence is repeated in entirety in the italicized example immediately following.”

Response: Thank you for the suggestion. The authors agree with the suggestion and deleted lines 1 to 3.

Suggestions: P15 Line 10 and 11. Insert “United Nations” before ‘Convention on the rights of PwDs…” Also, capitalize “R”ights of “P”ersons with “D”isabilities.

Response: Thank you for the suggestion. The authors agree and have inserted the United Nations accordingly. The sentence now reads “The Government of Tuvalu in 2014 ratified the United Nations Convention on the Rights of Persons with Disabilities.”

---

## [Decision Letter · Decision Letter 1]

12 Oct 2020

Exploring Persons with Disabilities Preparedness, Perceptions and Experiences of Natural Disasters in Tuvalu

PONE-D-20-18521R1

Dear Dr. Mohammadnezhad,

We’re pleased to inform you that your manuscript has been judged scientifically suitable for publication and will be formally accepted for publication once it meets all outstanding technical requirements.

Kind regards,

Stefano Federici, Ph.D.

Academic Editor

PLOS ONE

Additional Editor Comments (optional):

Reviewers' comments:

Reviewer's Responses to Questions

**Comments to the Author**

1. If the authors have adequately addressed your comments raised in a previous round of review and you feel that this manuscript is now acceptable for publication, you may indicate that here to bypass the “Comments to the Author” section, enter your conflict of interest statement in the “Confidential to Editor” section, and submit your "Accept" recommendation.

Reviewer #1: (No Response)

Reviewer #2: All comments have been addressed

Reviewer #3: All comments have been addressed

2. Is the manuscript technically sound, and do the data support the conclusions?

Reviewer #1: Yes

Reviewer #2: Yes

Reviewer #3: Yes

3. Has the statistical analysis been performed appropriately and rigorously? 

Reviewer #1: Yes

Reviewer #2: Yes

Reviewer #3: N/A

4. Have the authors made all data underlying the findings in their manuscript fully available?

Reviewer #1: Yes

Reviewer #2: Yes

Reviewer #3: Yes

5. Is the manuscript presented in an intelligible fashion and written in standard English?

Reviewer #1: Yes

Reviewer #2: Yes

Reviewer #3: Yes

6. Review Comments to the Author

Reviewer #1: Page 14 " cost the lives of PwDs in Sri Lanka as they did not understand the information to evacuate

during the 2015 Tsunami" & page 47 "during the 2015 Tsunami"- do you mean the 2005 Tsunami?

Table 3 is not core to results and could be relegated to an appendix and successive tables renumbered.

Some of the quotations contain grammatical errors. This is quite normal when interviewing respondents and reproducing their statements verbatim. However, you state that original responses were in Tuvaluan so errors are those of the transcription unless you were making attempts to translate grammatical errors accurately. This is highly unlikely as they would not translate directly.

I suggest you carry out a further in depth proof read of the whole paper to correct a small number of minor errors as it is not possible to annotate this pdf copy.

Reviewer #2: It is publishable manuscript. I would recommend to accept the paper. It seems to me that the author has revised the manuscript to address other reviewers' comments.

Reviewer #3: Reviewer 3. All suggestions from my earlier review addressed satisfactorily. Recommend acceptance. A very useful addition to small but growing body of important pasifika research on disability issues.

7. PLOS authors have the option to publish the peer review history of their article (what does this mean?). If published, this will include your full peer review and any attached files.

Reviewer #1: **Yes: **David King

Reviewer #2: No

Reviewer #3: No

---

## [Editor Report · Acceptance letter]

20 Oct 2020

PONE-D-20-18521R1 

Exploring Persons with Disabilities Preparedness, Perceptions and Experiences of Disasters in Tuvalu 

Dear Dr. Mohammadnezhad:

I'm pleased to inform you that your manuscript has been deemed suitable for publication in PLOS ONE. Congratulations! Your manuscript is now with our production department. 

Kind regards, 

on behalf of

Prof. Stefano Federici 

Academic Editor

PLOS ONE